# Validation of stearoyl-coA desaturase gene TaqMan probe-based SNP for genotyping Tattykeel Australian White MARGRA lamb for health-beneficial omega-3 long-chain fatty acids and intramuscular fat content

Aduli Enoch Othniel Malau-Aduli[1]*, Shedrach Benjamin Pewan[2],
Felista Waithira Mwangi[3], John Roger Otto[1]

1 School of Environmental and Life Sciences, College of Engineering, Science and Environment, The University of Newcastle, Callaghan, New South Wales, Australia, 2 National Veterinary Research Institute, Vom, Plateau State, Nigeria, 3 School of Medicine and Public Health, College of Health, Medicine and Wellbeing, The University of Newcastle, Callaghan, New South Wales, Australia

☾ These authors contributed equally to this work.
* aduli.malauaduli@newcastle.edu.au, aduli40@yahoo.co.uk

## Abstract

The Tattykeel Australian White (TAW) branded MARGRA lamb is a premium Australian sheep breed developed through decades of careful linebreeding, with the aim of improving the natural levels of health beneficial omega-3 long-chain polyunsaturated fatty acids (n-3 LC-PUFA), micro-marbled intramuscular fat (IMF) content, and low fat melting point (FMP) in the highly priced *Longissimus dorsi* muscle. These traits are key attributes contributing to the well documented superior meat eating quality of the TAW lamb, hence, fundamental to the TAW breeding program. The focus on Stearoyl-CoA Desaturase (SCD) gene is relevant because it plays an essential part in the biosynthesis of n-3 LC-PUFA and IMF metabolism. Previous studies had identified and established significant associations between the g.23881050T > C single nucleotide polymorphism (SNP) in the SCD gene and meat eating quality traits. Therefore, the aim of this study was to develop a reliable, fast, cost-effective, and accurate TaqMan probe-based Real-Time PCR assay for genotyping the identified SCD SNP in 118 TAW, Poll Dorset, Martin More sheep, and Angus bulls (as a distinguishing species control). By designing allele-specific TaqMan probes targeting the polymorphic locus, the assay accurately identified two genotypes, homozygous CC and heterozygous CT, at an estimated cost of AUD 2.16 and processing time of 49.30 minutes per 96-well sample plate. The genotype frequencies for CC and CT were 0.58 and 0.42 in rams, and 0.57 and 0.43 in ewes, respectively. The major and minor allele frequencies in TAW rams and ewes were 0.21 and 0.79, while in young TAW rams, they were 0.17 and 0.83, respectively. Both Poll Dorset and Martin More sheep exclusively displayed the CC genotype at a minor allele frequency of 1.00. The

**Data availability statement:** All relevant data are within the manuscript and its Supporting Information files.

**Funding:** This research was funded by the Science Industry Endowment Fund Ross Metcalf STEM Business Fellowship grant number G2400460, and co-funded by Tattykeel Australian White Pty Ltd. grant number G2300890. The funders had no part in the study design, data collection, analysis, or decision to publish, and manuscript preparation. 1. https://www.csiro.au/en/work-with-us/funding-programs/sme/stem-plus-business/sief-ross-metcalf-fellowship 2. https://www.tattykeel.com.au/.

**Competing interests:** The authors have declared that no competing interests exist.

findings validate the identified SNP g.23881050T > C as a suitable DNA marker for breeding for meat quality traits. and the validated TaqMan probe-based genotyping assay is a simple, reliable, fast, precise and cost-effective tool for genotyping TAW. Therefore, the developed and validated TaqMan probe herein, can facilitate genetic selection within the TAW breeding program, and assist in streamlining efforts to enhance meat eating quality of the breed.

## Introduction

Genetic selection is a reliable long-term strategy for improving meat omega-3 long-chain polyunsaturated fatty acids (n-3 LC-PUFA) profile and lamb eating quality in Tattykeel Australian White (TAW) sheep. However, evaluating meat quality traits is costly and mostly done post-mortem. Whole genome sequencing methods such as genome-wide association studies (GWAS), are expensive [1]. Predictive models like the Best Linear Unbiased Predictor (BLUP) are still limited by inaccuracies, insufficient validation and unreliable data [2,3]. In contrast, the emergence of targeted sequencing and next-generation sequencing platforms has allowed for precise identification of functional single nucleotide polymorphisms (SNP) in selected lipogenic genes associated with meat eating quality traits [4]. This has enabled the development of genetic markers for accurate, precise and efficient genetic selection of TAW lambs with superior meat eating quality. Therefore, the validation of identified SNP markers is essential for the continuous future breeding of TAW lambs with superior meat eating quality.

The ability of lamb producers to keep pace with the meat eating quality standards demanded by consumers is particularly important in Australia [5], USA [6], and Japan [7], where it directly affects market demand and price. The demand for superior quality meat with beneficial health-promoting properties is growing, with consumers and governing bodies increasingly focused on safe, healthy and nutritionally rich meat products [8]. Specifically, there is a rising interest in meat containing high levels of health-beneficial n-3 LC-PUFA. These include eicosapentaenoic acid (EPA), docosahexaenoic acid (DHA) and docosapentaenoic acid (DPA), which contribute immensely to improved nutrition and health outcomes in humans [9,10]. Key meat eating attributes like intramuscular fat (IMF) content or marbling, fat melting point (FMP), and n-3 LC-PUFA composition are critical to consumer satisfaction because they influence taste, tenderness, and flavour [11].

In an attempt to meet the consumer's burgeoning demand for superior meat eating quality, the Gilmore Family in Black Springs, Oberon, New South Wales, Australia, developed the TAW MARGRA lamb. The TAW sheep is an Australian first developed through rigorous linebreeding and selection of Van Rooy, Poll Dorset, Dorper and Texel breeds. MARGRA is a top meat eating quality lamb brand, marked by high micro-marbled IMF and health-beneficial n-3 LC-PUFA profiles, and low FMP [11]. Details of the nutrient composition analysis of eight MARGRA lamb cuts and minced patties from TAW sheep can be found in the Appendix.

These quality traits have driven international demand for the TAW MARGRA lamb around the world. However, the popularity of this breed has also brought along some risks of imitators and fraudsters. Therefore, to safeguard this premier breed and brand together with its unique quality attributes against potential risks, it is essential to establish an authentic and brand-specific DNA test based on SNP. Additionally, a SNP marker-assisted selection index will support continuous genetic selection for premium meat eating quality in TAW lambs [12]. This will allow breeders to maintain a consistent quality product without the need for invasive muscle biopsy procedure or slaughter for generating meat quality data.

SNP markers are widely used in livestock breeding programs to enhance traits such as IMF, FMP, and health-promoting n-3 PUFAs [13] because of their abundance in the genome [14]. Our previous research on Stearoyl-CoA Desaturase (SCD), Fatty acid binding protein 4 (FABP4), and Fatty acid synthase (FASN) lipogenic genes in TAW lambs identified SNP in SCD (g.23881050T>C), FASN (FASN g.12323864A>G) and FABP4 (g.62829478A>G) that were significantly associated with IMF, FMP and n-3 LC-PUFA DHA (C22:6n-3) and DPA (C22:5n-3) [4]. This finding led to the eventual development of a marker-assisted selection index for breeding TAW sheep for superior MARGRA lamb eating quality. In this current study, the aim is to validate this identified non-synonymous SCD SNP g.23881050T>C and develop a simple cost-effective, fast, accurate and sensitive Real-Time PCR TaqMan probe assay for rapid genotyping of TAW MARGRA lambs (Table 1). This assay targets T/C alleles associated with DPA, DHA, and IMF, offering an accessible, precise tool for selecting TAW lambs with superior meat eating quality traits.

Researchers have employed various molecular techniques to genotype individual animals by detecting allelic variations [15]. Traditional methods such as sequencing and restriction fragment length polymorphism (RFLP) analysis, were deemed accurate but time-consuming, and have been generally applied on small sample sizes [15]. The tetra-primer Amplification refractory mutation system (ARMS) polymerase chain reaction (PCR) technique has also been used for genotyping through banding pattern analysis, but it is complex, laborious, and less effective for large datasets [14]. Kompetitive Allele-Specific PCR (KASP) offers a more affordable option, but its sensitivity and allele discrimination accuracy are limited [16]. A more recent method, known as the RNase H2-dependent PCR (rhPCR) (17), shows potential in SNP genotyping, but still requires further testing for broader application [16]. In contrast, TaqMan SNP genotyping, utilizing Real-Time PCR with fluorescent probes, has become popular due to its high throughput [18], sensitivity and specificity [19] to different DNA quality levels. Despite its advantages, TaqMan genotyping demands precise probe design to ensure accurate allele detection [20]. This progression in SNP genotyping highlights the need to balance efficiency, accuracy, sensitivity, and practical application of each technique.

Therefore, the primary objective of the current study was to validate the identified SCD SNP g.23881050T>C by developing a precise, efficient, and cost-effective TaqMan probe-based assay that could rapidly and accurately genotype TAW MARGRA lambs sustainably and reliably. The assay is designed to genotype T/C alleles associated with health beneficial n-3 LC-PUFA and IMF content.

Table 1. Mean (±SE) comparisons between genotypes of SCD SNP g.23881050T>C for docosahexaenoic acid (DHA), docosapentaenoic acid (DPA) and intramuscular fat (IMF) content in Tattykeel Australian White (TAW) lambs.

**SCD SNP g.23881050T>C**

| Genotype | DHA[a] (mg/100g) | DPA[b] (mg/100g) | IMFc (%) | Reference |
|---|---|---|---|---|
| CC | 7.00±2.11 | 17.9±6.81 | 3.98±0.312 | [4] |
| CT | 7.64±2.09 | 19.4±6.74 | 4.39±0.287 | |
| TT | 11.00±2.34 | 27.1±3.26 | 5.43±0.516 | |

[a]DHA, docosahexaenoic acid.

[b]DPA, docosapentaenoic acid.

[c]IMF, intramuscular fat.

## Materials and methods

### Animal ethics

All the procedures in this study were approved by the University of Newcastle Animal Care and Ethics Committee (Permit No. A-2024–416) in adherence to the Australian Code for the Care and Use of Animals for Scientific Purposes. The reporting in the manuscript follows the recommendations of Kilkenny et al. [21] in the ARRIVE guidelines for animal research

### Animals and experimental design

A total of 118 animals was studied, including TAW MARGRA ewes (n = 68), mature rams (n = 12), and young rams (n = 23), Poll Dorset (n = 5) and Martin More (n = 5) rams that served as positive controls. Angus bulls (n = 5) were used as negative controls. All animals were raised at the Tattykeel Australian White stud farm in Black Springs, Oberon, New South Wales, Australia. They grazed on ryegrass pastures in separate paddocks. The average age of ewes was 14, rams (33), young rams (9), and Angus bulls (15) months. The average liveweights in mature rams, ewes, young rams, and Angus bulls were 40, 36, 20 and 550 kg, respectively. Experimental animals were randomly selected, and sample size appropriateness was confirmed through a *priori* G-Power analysis (95%, F = 2.5) to ensure statistical robustness.

### Blood collection and genomic DNA extraction

Blood samples were collected via jugular venipuncture into 10 mL EDTA vacutainers and stored at −80°C. Genomic DNA (gDNA) was extracted using NucleoSpin Blood Kits (Macherey-Nagel GmbH, Germany) supplied by Scientifix® Australia Pty Ltd. DNA quantity and quality were assessed using a NanoDrop™ One/OneC Microvolume UV-Vis Spectrophotometer (Thermo Fisher Scientific, Australia). Acceptable gDNA yield and purity were confirmed at A260/A280 ≥ 1.65 and A260/A230 ≥ 1.65.

### Primer design for SCD gene sequence

Forward and reverse primers for next generation sequencing of SCD gene was designed using Geneious Prime Software Program 2020 v.2.2 (http://www.geneious.com) and synthesised by Integrated DNA Technologies Pte. Ltd. Details are shown in Table 2.

### Custom TaqMan SNP assay design

The TaqMan SNP assay uses TaqMan 5′-nuclease chemistry to amplify and detect specific SNP in gDNA, therefore quality sequence is paramount to the successful outcome. Next-generation sequence data was utilised for the assay design. Pre-design criteria included:

- Target sequence length of 600 bases for optimal assay design.
- BLAST analysis to confirm sequence uniqueness and identify polymorphisms.
- Avoidance of repeat-rich regions to prevent non-specific amplification.

Table 2. Primer parameters of the SCD polymerase chain reaction assay.

| Primer | Sequence | Length | ªTa (ºC) | Fragment Length (bp) | Reference |
|--------|----------|--------|----------|----------------------|-----------|
| Forward | CAAACTTAGGTCTGCAACTTTCGT | 24 | 65 | 11,545 | [4] |
| Reverse | TTTCCCACTTCAACTCACCCTATT | 24 | 65 | | |

ªTa **(ºC)**, annealing temperature.

The primers and probes targeting the g.23881050T > C locus of the SCD lipogenic gene were designed and synthesized by ThermoFisher Scientific, Australia. The sequence-specific probes distinguished alternate alleles (Allele 1: C/C, VIC dye; Allele 2: T/T, FAM dye) and incorporated a minor groove binding nonfluorescent quencher (MGBNFQ) at the 3′-end. Intellectual property restrictions and agreement prevented disclosure of specific sequences of primers and probes in this paper, which are registered with government authorities by Tattykeel Pty Ltd.

**Real-time PCR (RT-PCR)**

Extracted gDNA was diluted to a working concentration of 20 ng/µL using TE buffer (10 mM Tris-HCl, 1 mM EDTA, pH 8.0; ThermoFisher Scientific, Australia). Concentrations were verified using the Quantifluor dsDNA System (Promega, USA). PCR reactions (10 µL) consisted of 5.0 µL TaqPath ProAmp Master Mix, 0.5 µL TaqMan SNP Genotyping Assay, and 4.5 µL gDNA or no-template control (NTC). The QuantStudio-3 Real-Time PCR System (Applied Biosystems) was used under fast-cycling conditions: 60°C for 30 seconds (pre-read), 95°C for 20 seconds (denaturation/enzyme activation), followed by 40 cycles of 95°C for 1 seconds (denaturation) and 60°C for 20 seconds (annealing/extension), with a final 60°C for 30 seconds (post-read).

**Genotype frequency calculation**

Total Population Size (N)
   The total population size was calculated as:

$$N = n(CC) + n(CT) + n(TT)$$

For each group, the genotype frequencies were calculated as:
   Genotype Frequency:
   Frequency of CC = n(CC)/ N
   Frequency of CT = n(CT)/ N
   Frequency of TT = n(TT)/ N

**Allele frequency calculation**

The total number of alleles in the population is given by:

$$Total\ Alleles = 2 \times N$$

Each genotype contributes alleles as follows:
   CC: 2 alleles of C per individual
   CT: 1 allele of C and 1 allele of T per individual
   TT: 2 alleles of T per individual
The frequencies of C and T were calculated as:

$$Frequency\ of\ C :\ 2 \times n(CC) + n(CT)/ 2 \times N$$

$$Frequency\ of\ T :\ 2 \times n(TT) + n(CT)/ 2 \times N$$

**Major and minor allele frequency calculations**

The Taqman SNP genotyping results based on observed and expected genotypes were used to compute major and minor allele frequencies using the Hardy–Weinberg equilibrium principle, as indicated below [22].

$$p^2 + 2pq + q^2 = 1$$

Where;

   $p^2$ is the frequency of CC homozygotes
   $2pq$ is the frequency of CT heterozygotes
   $q^2$ is the frequency of TT homozygotes

## Results

The TaqMan probe-based assay successfully genotyped the g.23881050T>C SNP within the SCD gene across TAW, Poll Dorset, and Martin More sheep populations. Notably, the TAW sheep exhibited two genotypes (CC and CT), while the Poll Dorset and Martin More sheep displayed exclusively the CC genotype (Table 3). As indicated in Fig 1 and 2, there were distinct genotype proportions displayed in in TAW sheep as shown in the amplification plot. Fig 2 shows that unlike the conventional clustering of three genotypes, Homozygous Allele 1/Allele 1, Heterozygous Allele 1/Allele 2, and Homozygous Allele 2/Allele 2, the TAW population showed only two clusters: Homozygous Allele 2/Allele 2 and Heterozygous Allele 1/Allele 2.

The genotype and allele frequencies of the SCD gene varied across groups and are summarized in Table 3. In the TAW population, genotype frequencies for CC and CT were 0.58 and 0.42 in rams, and 0.57 and 0.43 in ewes, respectively. In young TAW lambs, genotype frequencies differed slightly from the rams and ewes, with CC and CT observed at 0.65 and 0.35, respectively. Poll Dorset and Martin More exhibited a single genotype, CC, with a frequency of 1.00. Allele frequencies in TAW rams (C=0.79, T=0.21) and ewes (C=0.79, T=0.21) reflected a higher frequency for the C allele. This trend was even more pronounced in young TAW lambs (C=0.83, T=0.17). By contrast, all the Poll Dorset and Martin More sheep populations had the C allele (frequency=1.00). As expected, no genotype or allele frequencies were detected for the Angus bulls (Table 3).

The major and minor allele frequencies observed across the different breeds are shown in Table 4. The major allele frequencies for TAW ram and ewe were identical at 0.21, while the young TAW ram showed a lower major allele frequency of 0.17. In contrast, Poll Dorset, Martin More, and Angus bull all exhibited a major allele frequency of 0.00. The young rams had a minor allele frequency at 0.83, followed by the TAW ram and ewe at 0.79 each. Meanwhile, the minor allele frequencies for Poll Dorset and Martin More sheep were all at 1.00, whereas the Angus bull recorded a value of 0.00. These differences highlight breed-specific genetic variation.

Additionally, we calculated the cost-efficiency and time requirements of the TagMan SNP assay, detailed in Table 5. Running a single sample costs AUD 2.16, based on a 10 µL reaction volume in a 96-well plate. This cost includes the TagMan probes, primers, and 10 µL of TaqPath ProAmp master mix. The assay is time-efficient, requiring just 49.30 minutes for a 96 well-plate setup and PCR runtime.

**Table 3. SCD genotypes and allele frequencies in Tattykeel Australian White (TAW) rams, ewes, young rams, poll Dorset, Martin More and Angus bull.**

| Breed | Genotype | | | Genotype Frequency | | | Allele Frequency | |
|---|---|---|---|---|---|---|---|---|
| | TT | CT | CC | TT | CT | CC | T | C |
| **Poll Dorset** | 0 | 0 | 5 | 0 | 0.00 | 1.00 | 0.00 | 1.00 |
| **Martin More** | 0 | 0 | 5 | 0 | 0.00 | 1.00 | 0.00 | 1.00 |
| **TAW Rams** | 0 | 5 | 7 | 0 | 0.42 | 0.58 | 0.21 | 0.79 |
| **TAW Ewes** | 0 | 29 | 39 | 0 | 0.43 | 0.57 | 0.21 | 0.79 |
| **TAW Young** | 0 | 8 | 15 | 0 | 0.35 | 0.65 | 0.17 | 0.83 |
| **Angus Bull** | 0 | 0 | 0 | 0 | 0.00 | 0.00 | 0.00 | 0.00 |

**A**

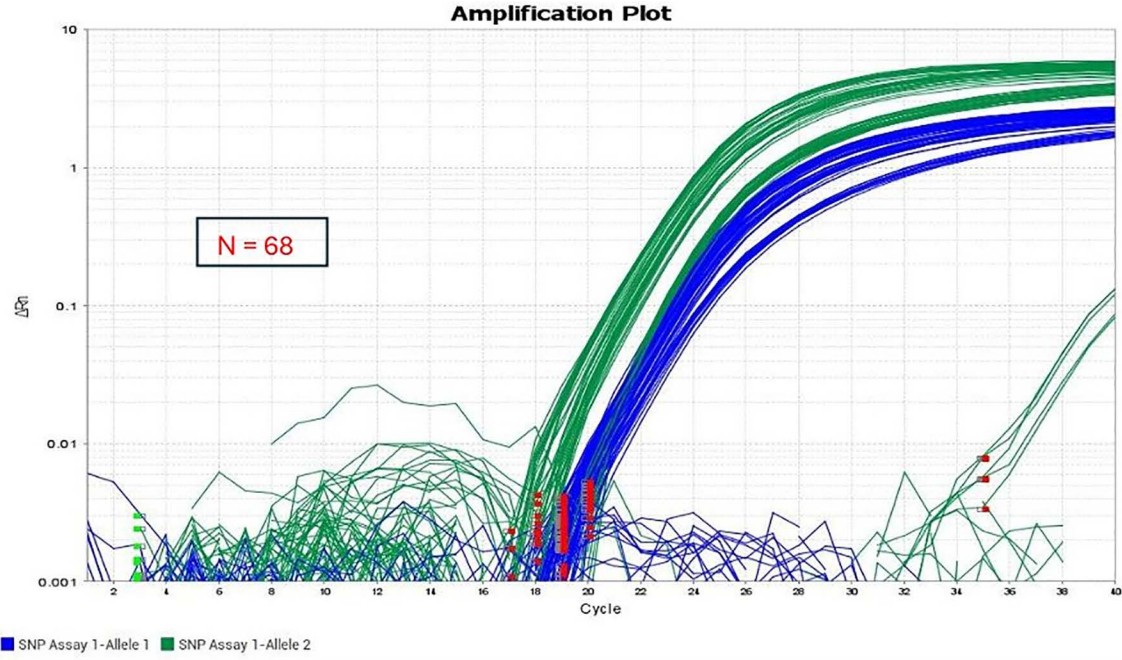

SNP Assay 1-Allele 1   SNP Assay 1-Allele 2

**B**

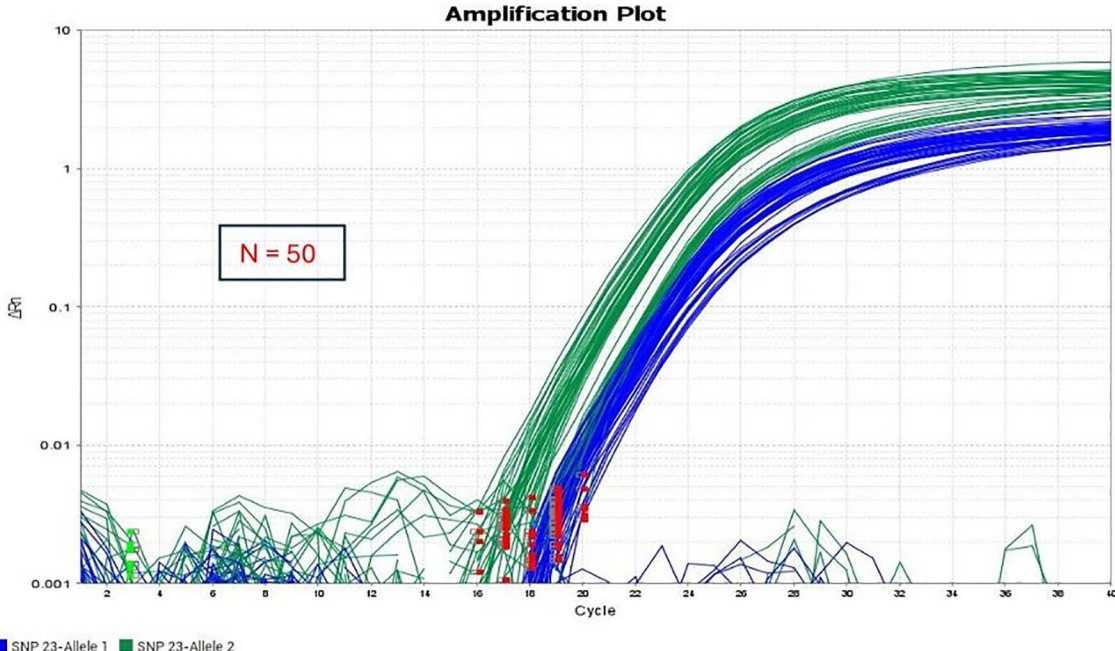

SNP 23-Allele 1   SNP 23-Allele 2

**Fig 1. The amplification curves illustrating the genotypic profiles of homozygotes and heterozygote for both alleles of the SCD gene in Tat-tykeel Australian White (TAW), Poll Dorset and Martin More sheep, and Angus bull populations, showing precision and consistency in genetic analysis of 118 samples (A, n = 68; B, n = 50).**

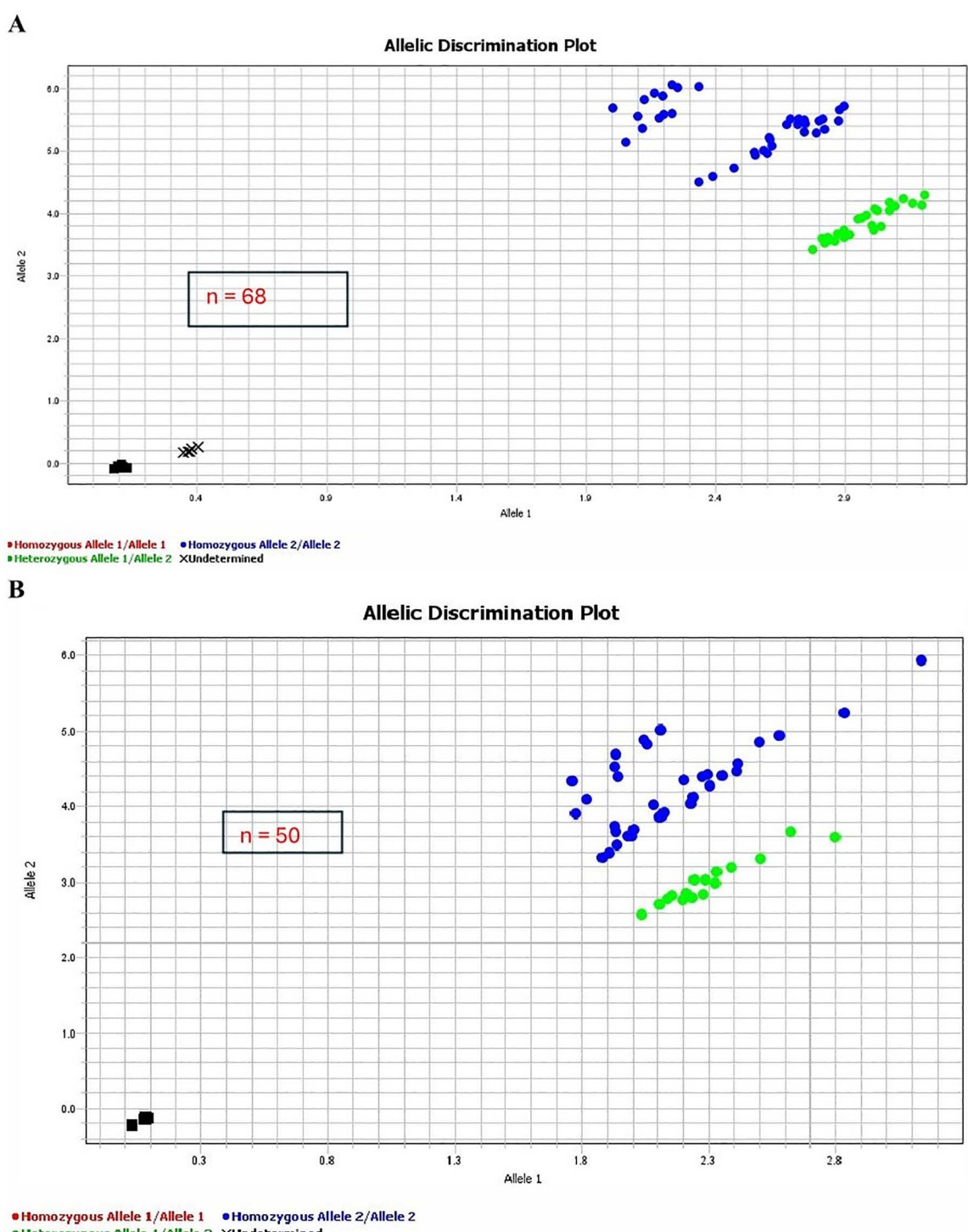

**Fig 2. Allelic discrimination plot for SNP g.23881050T > C in the SCD gene, generated using TaqMan genotyping assays on 118 samples from Tattykeel Australian White (TAW), Poll Dorset, Martin More, and Angus bull breeds (A, n = 68; B, n = 50). The blue dots indicate homozygous recessive genotype, green dots represent the heterozygous genotypes, black X corresponds to undetermined Angus bull samples, and black squares denote no-template controls.**

**Table 4. Major and minor allele frequencies of SCD gene SNP in Tattykeel Australian White (TAW), Poll Dorset, Martin More and Angus bull.**

| Breed | Genotype | | | Major Allele Frequency | Minor Allele Frequency |
|---|---|---|---|---|---|
| | TT | CT | CC | | |
| TAW Ram | 0 | 5 | 7 | 0.21 | 0.79 |
| TAW Ewe | 0 | 29 | 39 | 0.21 | 0.79 |
| TAW Young Ram | 0 | 8 | 15 | 0.17 | 0.83 |
| Poll Dorset | 0 | 0 | 5 | 0.00 | 1.00 |
| Martin More | 0 | 0 | 5 | 0.00 | 1.00 |
| Angus Bull | 0 | 0 | 0 | 0.00 | 0.00 |

**Table 5. TaqMan SNP assay cost per sample.**

| TaqMan | Cost[a] (AUD) | No. Samples | AUD/Sample | Total AUD/Sample |
|---|---|---|---|---|
| Assay (187 uL) | 590.65 | 374.00 | 1.58 | |
| Master mix (10 mL) | 1157.00 | 2000.00 | 0.58 | 2.16 |
| Time Required for 96-wells Plate Preparation | | | | |
| Item | Time (min) | | | |
| Plate preparation (96-well-plate) | 15.00 | | | |
| qPCR run time (Fast protocol) | 34.30 | | | |
| Total (min) | 49.30 | | | |

[a]AUD = Australian dollar.

## Discussion

In our previous study, we identified a novel SNP (g.23881050T > C) within the SCD gene locus in TAW sheep, which was linked to health-promoting n-3 LC-PUFAs (DHA and DPA) and IMF content [4]. Among the genotypes, TT exhibited the highest levels of DHA (11.00 ± 2.34 mg/100g), DPA (27.1 ± 3.26 mg/100g), and IMF (5.43 ± 0.516%), outperforming the CT genotype (7.64 ± 2.09 mg/100g, 19.4 ± 6.74 mg/100g, 4.39 ± 0.287%) and CC genotype (7.00 ± 2.11 mg/100g, 17.9 ± 6.81 mg/100g, 3.98 ± 0.312%), which showed median and lowest levels, respectively (Table 1). These findings provide a foundation for marker-assisted selection in TAW herds, aiming to enhance n-3 LC-PUFA profiles, increase IMF, and reduce FMP.

The discovery of this functional SCD SNP enabled the development of a precise TaqMan probe-based SNP real-time PCR assay at the core of this study. As consumer demand for healthy and nutritious red meat grows [23], breeding livestock with higher levels of n-3 LC-PUFA becomes increasingly important. The n-3 LC-PUFAs, DHA, DPA, and EPA are particularly known for their roles in reducing cardiovascular disease risk [24] and mitigating conditions such as cancer, inflammation, diabetes, and arthritis [25,26]. These fatty acids also contribute to bone and brain development in children [27].

SNP are now widely used in livestock genetic selection due to their abundance in the genome [28]. The SCD lipogenic gene, essential for fat metabolism in muscle tissue [29], has been extensively studied for its impact on n-3 LC-PUFA and IMF [13]. It plays a critical role in producing soft fats with low FMP [30] and improving meat eating quality traits such as tenderness, juiciness and flavour. The SCD gene encodes enzymes responsible for desaturating saturated fatty acids (SFA) by introducing *cis* double bonds at the *9th* and *10th* carbon positions [29,31]. Studies have linked SNP within this gene to n-3 LC-PUFA, IMF, FMP, and overall meat quality [32]. For instance, research on Wagyu x Limousin crossbreeds found SCD1 alleles had a positively relationship with marbling and monounsaturated fatty acids (MUFA) and inversely correlated with SFA levels [33].

The SNP 878 substitution (C to T) in the fifth exon of the SCD gene has distinct genotype effects. Studies on Wagyu [30] and Fleckvieh cattle [34] revealed that the AA genotype produced higher MUFA and IMF but lower FMP and SFA than the VA and VV genotypes. Similar associations between SCD SNP and fatty acid composition have been reported in sheep [35] and goats [36].

Our study confirmed the presence of CC and CT genotypes in the TAW sheep population, with CC being more prevalent. This validates the SNP marker's effectiveness in genetic selection for TAW MARGRA lambs. The TaqMan probe-based assay proved to be a reliable, precise, and high-specificity molecular tool for TAW genotyping. Costing AUD 2.16 per sample and requiring a total processing time of 49.30 minutes per 96-well plate, it offers an affordable and rapid genotyping solution for TAW breed.

The amplification plots for the g.23881050T > C SNP in all three sheep populations, shown in Fig 1, demonstrate quality and consistency, affirming the robust performance of the TaqMan probe-based SNP assay used in this experiment.

Traditionally, TaqMan discrimination plots show three genotype clusters along with a no-template control [16]. However, in this study, only two clusters were observed (Fig 2). Factors affecting cluster separation can include sample preparation, gDNA quality, instrumentation, qPCR settings [37], and complexities in SNP assay design [38]. To ensure accuracy, rigorous quality controls were implemented at every stage, from next-generation sequencing to SNP assay design and genotyping. Genomic DNA quality met stringent purity and yield criteria (A260/A280 ≥ 1.65 and A260/A230 ≥ 1.65). TaqMan probes and primers, synthesized by ThermoFisher experts in Australia, were thoroughly tested before use. Our laboratory routinely operates real-time PCR instrumentation for genotyping, ensuring consistency and accuracy. The next-generation sequencing data used to design this TaqMan assay had been previously published [4], thus reinforcing the study's credibility and validity.

In this study, TaqMan assay consistently identified two genotypes, CC and CT (Figs 1 and 2), aligning with findings from previous studies such as Lina et al. [39], which also reported two genotypes when genotyping the Myostatin gene in Chinese Tan sheep. This suggests that TaqMan assays may reveal limited genetic variation due to breed homogeneity. Intensive selective breeding programs targeting specific phenotypic traits have been linked to reduced genetic variability, leading to breed homogeneity [40].

Genetic selection aims to increase the frequency of desirable alleles for traits of interest [41]. The rare homozygous CC genotype was highly prevalent, suggesting that the pursuit of superior meat eating quality in TAW lambs through genetic selection has reduced genetic diversity to CC and CT genotypes. Remarkably, the TT genotype was absent from our sampled population. The loss of favourable alleles could be attributed to factors such as low linkage disequilibrium and genetic drift [41]. The Tattykeel Australian White sheep is a product of several generations of selective line breeding practices over a 15-year period of rigorous breeding, culling and selection of Poll Dorset, Dorper, Texel and Van Rooy rams and ewes with an extensive utilisation of embryo transfer, artificial insemination and natural mating [42]. Both exons and introns are known to contribute to gene and genome functionality where introns encode regulatory elements which participate in transcription, splicing and recombination events [43]. However, it is possible to observe low minor allele frequency [MAF] due to loss-of-function polymorphisms, and several have been reported in sheep [44–46]. For instance, in their evidence for multiple alleles effecting muscling and fatness at the *Ovine GDF8* locus in 12 breeds of sheep, Kijas et al. [45] reported a loss of function allele in the homozygous genotype (TT) in 109 out of 116 Australian Texels tested. This included sires sampled from 12 different producers and indicated that the mutant allele was near fixation within commercial Australian Texel flocks. Similarly, Kong et al., [47] investigated the association between SNP within the SLC27A6 gene and fatty acid content in Hu sheep and reported the detection of some rare SNP with very low MAF of less than 0.05 that were subsequently excluded. Similar low MAF have been reported in Colored Polish Merino sheep [43], Norwegian White sheep [48,49], New Zealand sheep [46], Chinese indigenous sheep populations of Mongolian, Kazakh, Tibetan, and Yunnan origins [50], South African sheep [51], North Caucasian sheep [52], Ethiopian sheep [53], Indian sheep [54], Iranian Mehraban sheep [55], and UK Texel sheep [56]. Therefore, the combination of selective line breeding and low MAF of the TT genotype in TAW sheep, could explain why the TT genotype wasn't visible in the current study.

Pewan et al. [42] conducted a comprehensive review of lipogenic genes (including the SCD) and their associations with genetic selection for meat quality. In terms of the functional mechanisms and relevance of the SCD gene variants underlying the relationship between this SNP and fatty acid metabolism and meat quality traits in ruminants, the SCD gene encodes for delta-9 desaturase enzyme, an iron-containing endoplasmic reticulum enzyme that catalyses a rate-limiting step in the conversion of saturated fatty acids into monounsaturated fatty acids in mammalian adipose cells [57]. The principal product of the desaturase enzyme is oleic acid, which is formed by the desaturation of stearic acid. Thus, the SCD enzyme is essential in the biosynthesis of monounsaturated fatty acids such as oleic (C18: 1n-9) and palmitoleic (C16: 1n-9) acids, formed after the addition of a double bond in the Δ9 position of their precursors, C18: 0 and C16: 0 fatty acids, respectively [58]. In sheep, Dervishi et al. [59] reported that grazing raises the quantities of conjugated linoleic acid, total polyunsaturated fatty acids and omega-3 fatty acids in lamb, which is a favourable and desirable option in line with health-beneficial human dietary guidelines [60].

Li and Dehkordi [61] identified functional genes associated with sheep growth and meat-related traits, genes overlapping with genomic variants, structural variations, allele and copy number variations associated with selection and breed improvement programmes in domestic and wild sheep. While selective breeding has successfully improved livestock traits, it can also lead to changes in allele frequencies with unintended consequences on genetic diversity. For instance, Liu et al. [62] investigated allele frequency changes in genomic selection programmes aimed at understanding the impact of using long-term genomic selection on changes in allele frequencies, genetic variation and level of inbreeding, and concluded that when implementing long-term genomic selection, strategies for genomic control of inbreeding are essential, due to a considerable hitch-hiking effect. Also, intensive selection in commercial broiler chickens enhanced growth rates and meat quality, but increased undesirable compounds like purines and uric acid [63]. Similarly, in dairy cattle, selection for higher milk production reduced genetic diversity in Estonian Holsteins [40]. These findings align with our allele frequency results, suggesting that breeding for specific traits in TAW MARGRA lambs altered allele frequencies that may potentially also reduce genetic diversity in the sampled population. Thus, targeted breeding should be carefully managed to prevent excessive genetic uniformity. Given our sample size of 118, future studies should expand the dataset to strengthen these findings. Nonetheless, this research underpins the potential of the TaqMan probe-based SNP real-time PCR assay as a valuable tool in TAW sheep breeding programs. By integrating this technology, TAW lamb producers can efficiently enhance meat quality, meet consumer demand for healthier red meat, and achieve cost-effective genetic improvements.

## Conclusion

The TaqMan SNP genotyping analysis of the SCD gene in TAW sheep revealed the presence of two distinct genotypes, CC and CT, at the g.23881050T > C locus. Furthermore, there were low major allele frequencies and high minor allele frequencies in all three TAW groups. These findings underpin the genetic uniqueness of TAW sheep compared to other breeds, offering valuable insights into their distinctive genetic makeup. By employing a TaqMan SNP genotyping assay, this study demonstrates the potential of a simple, cost-effective, and fast reliable tool for genotyping TAW using SCD SNP. This method not only delivers precise and high-quality results but also serves as an accessible and efficient solution that can be used in breeding programs and genetic research within the TAW sheep herd population. Moreover, its affordability and fast processing time make it an ideal choice for large-scale genotyping, opening the way for the genetic improvement and sustainable management of TAW sheep breed in Australia. However, due to the genetic selection for n-3 LC-PUFA, IMF and FMP meat eating quality traits, a potential breed homogeneity may occur in TAW herd population.

## Limitations

The following limitations of this study should be taken into account: First, the study genotyped a total of 118 animals only. Due to the relatively small sample size, this may not fully capture the genetic diversity of the entire TAW population, hence

further validation in larger and more diverse cohorts is warranted in future studies. Second, although the TaqMan assay technology is robust, effective and rapid, there are potential limitations associated with detecting rare alleles, allele drop-out and probe specificity that should be taken into consideration.

## Supporting information

**S1 Appendix. Genotyped data.**
(XLSX)

**S2 Appendix. Nutrient composition analysis of eight TAW MAGRA lamb cuts and minced patties.**
(DOCX)

## Acknowledgments

The authors gratefully acknowledge The Gilmores of Tattykeel Australian White Pty Ltd, Oberon, New South Wales, Australia for access to herd and farm resources, and the Commonwealth Scientific and Industrial Research Organization SIEF Ross Metcalf STEM Business Industrial Research Fellowship programme. We would also like to appreciate the support of the School of Environmental and Life Sciences, College of Engineering, Science and Environment, The University of Newcastle, Callaghan, New South Wales, Australia.

## Author contributions

**Conceptualization:** Aduli Enoch Othniel Malau-Aduli.

**Data curation:** Shedrach Benjamin Pewan, Felista Waithira Mwangi, John Roger Otto.

**Formal analysis:** Felista Waithira Mwangi, John Roger Otto.

**Funding acquisition:** Aduli Enoch Othniel Malau-Aduli.

**Investigation:** Shedrach Benjamin Pewan, Felista Waithira Mwangi, John Roger Otto.

**Methodology:** Aduli Enoch Othniel Malau-Aduli, Felista Waithira Mwangi, John Roger Otto.

**Project administration:** Aduli Enoch Othniel Malau-Aduli.

**Resources:** Aduli Enoch Othniel Malau-Aduli.

**Software:** Aduli Enoch Othniel Malau-Aduli.

**Supervision:** Aduli Enoch Othniel Malau-Aduli.

**Validation:** Aduli Enoch Othniel Malau-Aduli.

**Visualization:** Aduli Enoch Othniel Malau-Aduli, John Roger Otto.

**Writing – original draft:** John Roger Otto.

**Writing – review & editing:** Aduli Enoch Othniel Malau-Aduli, Shedrach Benjamin Pewan, Felista Waithira Mwangi.

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
