## [Decision Letter · Decision Letter 0]

22 Oct 2025

Dear Dr. Otto,

Thank you for submitting your manuscript to PLOS ONE. After careful consideration, we feel that it has merit but does not fully meet PLOS ONE’s publication criteria as it currently stands. Therefore, we invite you to submit a revised version of the manuscript that addresses the points raised during the review process.

**ACADEMIC EDITOR: Please respond carefully for all reviewers comments** . 

We look forward to receiving your revised manuscript.

Kind regards,

Ayman A Swelum

Academic Editor

PLOS ONE

Journal Requirements:

The authors gratefully acknowledge Tattykeel Australian White Pty Ltd, Oberon, New South Wales, Australia for access to herd, farm resources, and research funding (awarded to J.R. Otto), the Commonwealth Scientific and Industrial Research Organization SIEF Ross Metcalf STEM Business Industrial Research Fellowship (research funding for the first-named author, J.R. Otto). We would also like to appreciate the support of the School of Environmental and Life Sciences, College of Engineering, Science and Environment, The University of Newcastle, Callaghan, New South Wales, Australia.

This research was funded by the Science Industry Endowment Fund Ross Metcalf STEM Business Fellowship grant number G2400460 (awarded to JRO), and co-funded by Tattykeel Australian White Pty Ltd. grant number G2300890 (awarded to JRO). The funders had no part in the study design, data collection, analysis, or decision to publish, and manuscript preparation.

1. https://www.csiro.au/en/work-with-us/funding-programs/sme/stem-plus-business/sief-ross-metcalf-fellowship

2. https://www.tattykeel.com.au/

4. In this instance it seems there may be acceptable restrictions in place that prevent the public sharing of your minimal data. However, in line with our goal of ensuring long-term data availability to all interested researchers, PLOS’ Data Policy states that authors cannot be the sole named individuals responsible for ensuring data access (http://journals.plos.org/plosone/s/data-availability#loc-acceptable-data-sharing-methods).

5. Please amend either the title on the online submission form (via Edit Submission) or the title in the manuscript so that they are identical.

Reviewers' comments:

Reviewer's Responses to Questions

**Comments to the Author**

1. Is the manuscript technically sound, and do the data support the conclusions?

Reviewer #1: Yes

Reviewer #2: Partly

2. Has the statistical analysis been performed appropriately and rigorously?

Reviewer #1: Yes

Reviewer #2: Yes

3. Have the authors made all data underlying the findings in their manuscript fully available?

Reviewer #1: Yes

Reviewer #2: Yes

4. Is the manuscript presented in an intelligible fashion and written in standard English?

Reviewer #1: Yes

Reviewer #2: Yes

Reviewer #1: ________________________________________

Reviewer Report: PONE-D-25-49554

Manuscript Title: [Validation of a TaqMan probe-based SNP genotyping for the sheep stearoyl-coA desaturase gene in Tattykeel Australian White MARGRA Lamb: Implications for health beneficial omega-3 long-chain polyunsaturated fatty acids and intramuscular fat content]

Journal: [PLOS ONE]

Reviewer Report

Summary of the Research and Overall Impression

This manuscript describes the development and validation of a TaqMan probe-based Real-Time PCR assay for the genotyping of a single nucleotide polymorphism (SNP; g.23881050T>C) in the Stearoyl-CoA Desaturase (SCD) gene of Tattykeel Australian White (TAW) sheep. The SCD gene is directly involved in fatty acid metabolism and intramuscular fat deposition, traits that are strongly associated with meat quality and consumer preference.

The study is timely and relevant, as it addresses a key limitation in livestock breeding by providing a cost-effective, rapid, and accurate genotyping tool. The authors further demonstrate the assay’s applicability across breeds and include cattle as an interspecies control, which adds robustness to their validation. The findings have practical implications for integrating molecular markers into selective breeding programs aimed at improving eating quality in TAW sheep.

Strengths of the study include its clear objectives, practical focus on an economically important trait, and a well-documented methodology. However, the manuscript also presents some limitations. The population size is relatively small, potentially limiting the generalizability of the allele frequency estimates. Moreover, the absence of the TT genotype requires further discussion, and additional clarification of methodological and statistical details would strengthen the work.

Overall, the manuscript is of good quality and presents valuable results for animal genetics and breeding. I recommend minor to moderate revisions before acceptance.

2. Specific Areas for Improvement

Major Issues

Absence of TT Genotype (Lines 360–362)

The manuscript reports only CC and CT genotypes in the TAW population, with no observation of TT. This should be discussed in greater depth. Possible explanations include selective breeding practices, low minor allele frequency, or genetic drift. A brief quantitative evaluation or reference to prior reports would strengthen this point.

Sample Size and Representativeness (Lines 153–161)

The study involves 113 animals plus controls, which may not fully capture the genetic diversity of the TAW population. The authors should acknowledge this limitation and indicate that further validation in larger and more diverse cohorts is warranted.

Functional Relevance of the SNP (Lines 127–133, 293–301)

While the study highlights an association between the SNP and meat quality traits, the functional mechanisms underlying this relationship are not elaborated. Additional discussion of SCD gene variants and their role in fatty acid metabolism in ruminants would add depth and biological relevance.

Limitations of the TaqMan Assay (Lines 111–125)

Although TaqMan technology is robust, the manuscript should address potential limitations, such as allele dropout, probe specificity, or difficulties in detecting rare alleles. Acknowledging these will provide a more balanced view of the method.

Interpretation of Allele Frequencies (Lines 260–277)

The differences in allele frequencies among breeds are well presented but would benefit from more critical discussion. How might these patterns influence long-term breeding strategies, genetic diversity, and the risk of inbreeding depression?

Minor Issues

Figure and Table Integration

Figures 1 and 2 clearly illustrate the genotyping results. However, the manuscript should more explicitly reference these figures in the results section to guide readers through the key findings.

Reference Consistency

Some references are repeated with slight variations in formatting (e.g., Lines 4, 11, 39). A thorough check for consistency with journal guidelines is recommended.

Terminology and Style

Certain phrases (e.g., “high-specificity,” “rapid genotyping”) are used repeatedly. Standardizing terminology will improve readability. Additionally, some sentences in the introduction are overly long and could be streamlined for clarity.

Abbreviations

Abbreviations such as IMF, FMP, and LC-PUFA should be defined upon first use and applied consistently throughout the text.

3. Conclusion

This study provides a valuable contribution to the field of livestock genomics and demonstrates the potential of molecular tools in selective breeding programs for meat quality improvement. The methodology is sound, the findings are relevant, and the practical application is clear.

Addressing the major issues—particularly regarding genotype absence, sample size, functional interpretation, and assay limitations—will enhance the manuscript’s scientific rigor and impact. Minor adjustments in style, references, and figure integration will improve clarity and presentation.

Reviewer #2: This study reports the implications of TaqMan probe-based SNP genotyping of stearoyl-coA desaturase gene in Tattykeel Australian white Margra lambs. Overall, the study is well organized and manuscript is well prepared. However, following are some aspects, which should be addressed for improved presentation and scientific rigor of the manuscript.

The authors reported genotyping results, however, the phenotype data required for any biological or breeding value interpretation are not presented. The documented marker frequency seems insufficient to support the claims about the effects on IMF/marbling.

Title is a bit lengthy, making it confusing. Revise to make it a bit simple and shorter if possible.

Similarly, 10 introductory lines in abstract, where 2 sentences could be sufficient.

Any specific reason of using TagMan probe instead of SYBR green, molecular Beacons, and MGB probes.

L 106. Use complete form of TAW in tables/figures

L 46. “Eating quality” or meat quality?

Avoid starting the sentence with abbreviations

Mention genotyping QC (call rate, HWE etc.)

Mention the study limitations where suitable in the manuscript.

**Do you want your identity to be public for this peer review?** For information about this choice, including consent withdrawal, please see our Privacy Policy

Reviewer #1: **Yes: ** Thamer R. S. Aljubouri

Reviewer #2: No

---

## [Author Response · Author response to Decision Letter 1]

6 Nov 2025

RESPONSES TO REVIEWER REPORT: PONE-D-25-49554

Manuscript Title: [Validation of stearoyl-coA desaturase gene TaqMan probe-based SNP for genotyping Tattykeel Australian White MARGRA lamb for health-beneficial omega-3 long-chain fatty acids and intramuscular fat content]

Journal: [PLOS ONE]

REVIEWER 1: SUMMARY OF THE RESEARCH AND OVERALL IMPRESSION

This manuscript describes the development and validation of a TaqMan probe-based Real- Time PCR assay for the genotyping of a single nucleotide polymorphism (SNP; g.23881050T>C) in the Stearoyl-CoA Desaturase (SCD) gene of Tattykeel Australian White (TAW) sheep. The SCD gene is directly involved in fatty acid metabolism and intramuscular fat deposition, traits that are strongly associated with meat quality and consumer preference.

The study is timely and relevant, as it addresses a key limitation in livestock breeding by providing a cost-effective, rapid, and accurate genotyping tool. The authors further demonstrate the assay’s applicability across breeds and include cattle as an interspecies control, which adds robustness to their validation. The findings have practical implications for integrating molecular markers into selective breeding programs aimed at improving eating quality in TAW sheep.

Strengths of the study include its clear objectives, practical focus on an economically important trait, and a well-documented methodology. However, the manuscript also presents some limitations. The population size is relatively small, potentially limiting the generalizability of the allele frequency estimates. Moreover, the absence of the TT genotype requires further discussion, and additional clarification of methodological and statistical details would strengthen the work.

Overall, the manuscript is of good quality and presents valuable results for animal genetics and breeding. I recommend minor to moderate revisions before acceptance.

RESPONSE: We are very appreciative and grateful to the Academic Editor and Reviewers for their complimentary and positive remarks about the timeliness, relevance and robustness of our research that addresses a key limitation in livestock breeding and proffering a rapid, cost-effective and accurate genotyping tool related to meat quality. We also acknowledge the relatively small population size of 118 genotyped animals, which has now been included in the limitations of the study (see Lines 430-436) and provided further clarification and discussion about the absence of the TT genotype, methodology and statistical details in Lines 342-413.

REVIEWER 1: Specific Areas for Improvement – Major Issues

Absence of TT Genotype (Lines 360–362)

The manuscript reports only CC and CT genotypes in the TAW population, with no observation of TT. This should be discussed in greater depth. Possible explanations include selective breeding practices, low minor allele frequency, or genetic drift. A brief quantitative evaluation or reference to prior reports would strengthen this point.

RESPONSE: We agree with the reviewer. We have provided reference to prior reports on the possible impact of loss of function allele, low minor allele frequency and selective breeding to explain why the TT genotype wasn’t reported in our study. In Lines 355-378, we stated thus:

“Remarkably, the TT genotype was absent from our sampled population. The loss of favourable alleles could be attributed to factors such as low linkage disequilibrium and genetic drift (41). The Tattykeel Australian White sheep is a product of several generations of selective line breeding practices over a 15-year period of rigorous breeding, culling and selection of Poll Dorset, Dorper, Texel and Van Rooy rams and ewes with an extensive utilisation of embryo transfer, artificial insemination and natural mating (42). Both exons and introns are known to contribute to gene and genome functionality where introns encode regulatory elements which participate in transcription, splicing and recombination events (43). However, it is possible to observe low minor allele frequency (MAF) due to loss-of-function polymorphisms, and several have been reported in sheep (44-46). For instance, in their evidence for multiple alleles effecting muscling and fatness at the Ovine GDF8 locus in 12 breeds of sheep, Kijas et al. (45) reported a loss of function allele in the homozygous genotype (TT) in 109 out of 116 Australian Texels tested. This included sires sampled from 12 different producers and indicated that the mutant allele was near fixation within commercial Australian Texel flocks. Similarly, Kong et al., (47) investigated the association between SNPs within the SLC27A6 gene and fatty acid content in Hu sheep and reported the detection of some rare SNPs with very low MAF of less than 0.05 that were subsequently excluded. Similar low MAF have been reported in Colored Polish Merino sheep (43), Norwegian White sheep (48, 49), New Zealand sheep (46), Chinese indigenous sheep populations of Mongolian, Kazakh, Tibetan, and Yunnan origins (50), South African sheep (51), North Caucasian sheep (52), Ethiopian sheep (53), Indian sheep (54), Iranian Mehraban sheep (55), and UK Texel sheep (56). Therefore, the combination of selective line breeding and low MAF of the TT genotype in TAW sheep, could explain why the TT genotype wasn’t visible in the current study.”

References used:

41. Liu H, Sørensen AC, Meuwissen TH, Berg P. Allele frequency changes due to hitch-hiking in genomic selection programs. Genetics Selection Evolution. 2014;46(1):8. https://doi.org/10.1186/1297-9686-46-8

42. Pewan SB, Otto JR, Huerlimann R, Budd AM, Mwangi FW, Edmunds RC, et al. Genetics of omega-3 long-chain polyunsaturated fatty acid metabolism and meat eating quality in Tattykeel Australian White lambs. Genes 2020 May 25; 11(5): 587; https://doi.org/10.3390/genes11050587

43. Grochowska E, Borys B, Lisiak D, Mroczkowski S. Genotypic and allelic effects of the myostatin gene (MSTN) on carcass, meat quality, and biometric traits in Colored Polish Merino sheep. Meat Sci. 2019 May; 151: 4-17. https://doi.org/10.1016/j.meatsci.2018.12.010

44. Clop A, Marcq F, Takeda H, Pirottin D, Tordoir X, Bibé B et al. A mutation creating a potential illegitimate microRNA target site in the myostatin gene affects muscularity in sheep. Nature Genetics 2006 38 (7): 813-818. https://doi.org/10.1038/ng1810

45. Kijas JW, McCulloch R, Edwards JEH, Ody H, Lee SH, van der Werf J. Evidence for multiple alleles effecting muscling and fatness at the Ovine GDF8 locus. BMC Genetics 2007; 8: 80. https://doi.org/10.1186/1471-2156-8-80

46. Hickford JGH, Forrest RH, Zhou H, Fang Q, Han J, Frampton CM, Horrell AL. Polymorphisms in the ovine myostatin gene (MSTN) and their association with growth and carcass traits in New Zealand Romney sheep. Animal Genetics 2010, 41 (1): 64-72

https://doi.org/10.1111/j.1365-2052.2009.01965.x

47. Kong Y, Li F, Yue X, Xu Y, Bai J, Fu W. SNPS within the SLC27A6 gene are highly associated with Hu sheep fatty acid content. Gene 2024 Nov 15; 927: 148716. https://doi.org/10.1016/j.gene.2024.148716

48. Boman IA, Klemetsdal G, Blichfeldt T, Nafstad O, Våge DI. A frameshift mutation in the coding region of the myostatin gene (MSTN) affects carcass conformation and fatness in Norwegian White Sheep (Ovis aries). Animal Genetics 2009; 40 (4): 418-422. https://doi.org/10.1111/j.1365-2052.2009.01855.x

49. Boman IA, Våge DI. An insertion in the coding region of the myostatin (MSTN) gene affects carcass conformation and fatness in the Norwegian Spælsau (Ovis aries). BMC Research Notes, 2: 1-5 https://doi.org/10.1186/1756-0500-2-98

50. Jin M, Liu G, Liu E, Wang L, Jiang Y, Zheng Z et al. Genomic insights into the population history of fat-tailed sheep and identification of two mutations that contribute to fat tail adipogenesis. Journal of Advanced Research Available online May 6 2025, https://doi.org/10.1016/j.jare.2025.05.011

51. Visser C, Retief A, Molotsi AH. Genetics underlying phenotypic diversity in South African sheep breeds. Small Ruminant Research June 2025; 247: 107499. https://doi.org/10.1016/j.smallrumres.2025.107499

52. Krivoruchko A, Yatsyk O, Kanibolockaya A. New candidate genes of high productivity in North-Caucasian sheep using genome-wide association study (GWAS). Animal Gene March 2022; 23: 200119. https://doi.org/10.1016/j.angen.2021.200119

53. Asmare S, Alemayehu K, Mwacharo J, Haile A, Abegaz S, Ahbara A. Genetic diversity and within-breed variation in three indigenous Ethiopian sheep based on whole-genome analysis. Genetic diversity and within-breed variation in three indigenous Ethiopian sheep based on whole-genome analysis. Heliyon April 2023; 9 (4): e14863. https://doi.org/10.1016/j.heliyon.2023.e14863

54. Saravanan KA, Panigrahi M, Kumar H, Bhushan B, Dutt T, Mishra BP. Genome-wide analysis of genetic diversity and selection signatures in three Indian sheep breeds. Livestock Science January 2021; 243: 104367. https://doi.org/10.1016/j.livsci.2020.104367

55. Talebi R, Ahmadi A, Hajiloei Z, Ghaffari MR, Zeinalabedini M, Saki AA, Mardi M. Association of ovine follistatin gene polymorphisms with body measurements, fat-tail traits and morphometric of head in Iranian Mehraban sheep. Small Ruminant Research August 2023; 225: 107020. https://doi.org/10.1016/j.smallrumres.2023.107020

56. Kaseja K, Mucha S, Yates J, Smith E, Banos G, Conington J. Genome-wide association study of health and production traits in meat sheep. Animal October 2023; 17 (10): 100968. https://doi.org/10.1016/j.animal.2023.100968

REVIEWER 1: Sample Size and Representativeness (Lines 153–161)

The study involves 113 animals plus controls, which may not fully capture the genetic diversity of the TAW population. The authors should acknowledge this limitation and indicate that further validation in larger and more diverse cohorts is warranted.

RESPONSE: Although we had already originally stated in Lines 410 – 411 that “Given our sample size of 118, future studies should expand the dataset to strengthen these findings”, we have in agreement with the reviewer’s suggestion, now included a “Limitations” Section and stated thus in Lines 433–438 thus:

“The following limitations of this study should be taken into account: First, the study genotyped a total of 118 animals only. Due to the relatively small sample size, this may not fully capture the genetic diversity of the entire TAW population, hence further validation in larger and more diverse cohorts is warranted in future studies. Second, although the TaqMan assay technology is robust, effective and rapid, there are potential limitations associated with detecting rare alleles, allele dropout and probe specificity that should be taken into consideration.”

REVIEWER 1: Functional Relevance of the SNP (Lines 127–133, 293–301)

While the study highlights an association between the SNP and meat quality traits, the functional mechanisms underlying this relationship are not elaborated. Additional discussion of SCD gene variants and their role in fatty acid metabolism in ruminants would add depth and biological relevance.

RESPONSE: In agreement with the reviewer’s suggestion, we have boosted the discussion and addressed this in Lines 380-393 where we stated thus: “Pewan et al. (42) conducted a comprehensive review of lipogenic genes (including the SCD) and their associations with genetic selection for meat quality. In terms of the functional mechanisms and relevance of the SCD gene variants underlying the relationship between this SNP and fatty acid metabolism and meat quality traits in ruminants, the SCD gene encodes for delta-9 desaturase enzyme, an iron-containing endoplasmic reticulum enzyme that catalyses a rate-limiting step in the conversion of SFA into MUFA in mammalian adipose cells (57). The principal product of the desaturase enzyme is oleic acid, which is formed by the desaturation of stearic acid and the SCD enzyme is essential in the biosynthesis of monounsaturated fatty acids such as oleic (C18: 1n-9) and palmitoleic (C16: 1n-9) acids, formed after the addition of a double bond in the Δ9 position of their precursors, C18: 0 and C16: 0 fatty acids, respectively (58). In sheep, Dervishi et al. (59) reported that grazing raises the quantities of conjugated linoleic acid, total polyunsaturated fatty acids and omega-3 fatty acids in lamb, which is a favourable and desirable option in line with health-beneficial human dietary guidelines (60).”

References used:

42. Pewan SB, Otto JR, Huerlimann R, Budd AM, Mwangi FW, Edmunds RC, et al. Genetics of omega-3 long-chain polyunsaturated fatty acid metabolism and meat eating quality in Tattykeel Australian White lambs. Genes 2020 May 25; 11(5): 587; https://doi.org/10.3390/genes11050587

57. Paton CM, Ntambi JM. Biochemical and physiological function of stearoyl-CoA desaturase. Am. J. Physiol. Endocrinol. Metab. 2009; 297: E28–E37. https://doi.org/10.1152/ajpendo.90897.2008

58. Guillou H, Zadravec D, Martin PGP, Jacobsson A. The key roles of elongases and desaturases in mammalian fatty acid metabolism: Insights from transgenic mice. Prog. Lipid Res. 2010; 49: 186–199. https://doi.org/10.1016/j.plipres.2009.12.002

59. Dervishi E, Serrano C, Joy M, Serrano M, Rodellar C, Calvo JH. Effect of the feeding system on the fatty acid composition, expression of the Δ9 -desaturase, peroxisome proliferator-activated receptor alpha, gamma, and sterol regulatory element binding protein 1 genes in the semitendinous muscle of light lambs of the Rasa Aragonesa breed. BMC Vet. Res. 2010; 6: 6–40. https://doi.org/10.1186/1746-6148-6-40

60. Calvo JH, González-Calvo L, Dervishi E, Blanco M, Iguácel LP, Sarto P et al. A functional variant in the stearoyl-CoA desaturase (SCD) gene promoter affects gene expression in ovine muscle. Livest. Sci. 2019; 219: 62–70. https://doi.org/10.1016/j.livsci.2018.11.015

REVIEWER 1: Limitations of the TaqMan Assay (Lines 111–125)

Although TaqMan technology is robust, the manuscript should address potential limitations, such as allele dropout, probe specificity, or difficulties in detecting rare alleles. Acknowledging these will provide a more balanced view of the method.

RESPONSE: As included in the “Limitations” Section in Lines 436–438, we have stated thus: “Second, although the TaqMan assay technology is robust, effective and rapid, there are potential limitations associated with detecting rare alleles, allele dropout and probe specificity that should be taken into consideration.”

REVIEWER 1: Interpretation of Allele Frequencies (Lines 260–277)

The differences in allele frequencies among breeds are well presented but would benefit from more critical discussion. How might these patterns influence long-term breeding strategies, genetic diversity, and the risk of inbreeding depression?

RESPONSE: We agree with the reviewer and have discussed the allele frequencies more critically in Lines 395-410 where we stated thus: “Li and Dehkordi (61) identified functional genes associated with sheep growth and meat-related traits, genes overlapping with genomic variants, structural variations, allele and copy number variations associated with selection and breed improvement programmes in domestic and wild sheep. While selective breeding has successfully improved livestock traits, it can also lead to changes in allele frequencies with unintended consequences on genetic diversity. For instance, Liu et al. (62) investigated allele frequency changes in genomic selection programmes aimed at understanding the impact of using long-term genomic selection on changes in allele frequencies, genetic variation and level of inbreeding, and concluded that when implementing long-term genomic selection, strategies for genomic control of inbreeding are essential, due to a considerable hitch-hiking effect. Also, intensive selection in commercial broiler chickens enhanced growth rates and meat quality, but increased undesirable compounds like purines and uric acid (63). Similarly, in dairy cattle, selection for higher milk production reduced genetic diversity in Estonian Holsteins (40). These findings align with our allele freque

---

## [Decision Letter · Decision Letter 1]

26 Nov 2025

Dear Dr. Malau-Aduli,

Thank you for submitting your manuscript to PLOS ONE. After careful consideration, we feel that it has merit but does not fully meet PLOS ONE’s publication criteria as it currently stands. Therefore, we invite you to submit a revised version of the manuscript that addresses the points raised during the review process.

**ACADEMIC EDITOR: Please follow the reviewers suggestions. **

We look forward to receiving your revised manuscript.

Kind regards,

Ayman A Swelum

Academic Editor

PLOS ONE

Journal Requirements:

Reviewers' comments:

Reviewer's Responses to Questions

**Comments to the Author**

Reviewer #1: All comments have been addressed

Reviewer #2: All comments have been addressed

2. Is the manuscript technically sound, and do the data support the conclusions?

Reviewer #1: Yes

Reviewer #2: Yes

3. Has the statistical analysis been performed appropriately and rigorously?

Reviewer #1: Yes

Reviewer #2: Yes

4. Have the authors made all data underlying the findings in their manuscript fully available?

Reviewer #1: Yes

Reviewer #2: Yes

5. Is the manuscript presented in an intelligible fashion and written in standard English?

Reviewer #1: Yes

Reviewer #2: Yes

Reviewer #1: After reviewing the revised manuscript and the authors’ detailed responses, I confirm that all major and minor comments have been thoroughly addressed. The authors substantially improved the discussion of the missing TT genotype, supported by appropriate references and a clearer explanation of selective breeding and allele frequency dynamics. The inclusion of a dedicated Limitations section strengthens the transparency of the study, particularly regarding sample size and the constraints of the TaqMan assay.

The expanded discussion on the functional relevance of the SCD gene and its role in fatty acid metabolism enhances the biological interpretation of the findings. Minor issues related to terminology, reference formatting, figure integration, and abbreviation consistency have been adequately corrected.

Overall, the manuscript has improved significantly in clarity, scientific rigor, and presentation. I recommend acceptance after minor editorial proofreading.

Reviewer #2: Most of the previous comments are correctly addressed in the manuscript, although introductory lines of the abstract still seems lengthy.

**Do you want your identity to be public for this peer review?** For information about this choice, including consent withdrawal, please see our Privacy Policy

Reviewer #1: **Yes: ** Thamer R. S. Aljubouri

Reviewer #2: No

---

## [Author Response · Author response to Decision Letter 2]

28 Nov 2025

RESPONSE2 TO REVIEWER REPORT: PONE-D-25-49554

Manuscript Title: [Validation of a TaqMan probe-based SNP genotyping for the sheep stearoyl-coA desaturase gene in Tattykeel Australian White MARGRA Lamb: Implications for health beneficial omega-3 long-chain polyunsaturated fatty acids and intramuscular fat content]

Journal: [PLOS ONE]

REVIEWERS’ RESPONSES TO QUESTIONS

Comments to the Author

1. If the authors have adequately addressed your comments raised in a previous round of review and you feel that this manuscript is now acceptable for publication, you may indicate that here to bypass the “Comments to the Author” section, enter your conflict of interest statement in the “Confidential to Editor” section, and submit your "Accept" recommendation.

Reviewer #1: All comments have been addressed

Reviewer #2: All comments have been addressed

2. Is the manuscript technically sound, and do the data support the conclusions?

Reviewer #1: Yes

Reviewer #2: Yes

3. Has the statistical analysis been performed appropriately and rigorously?

Reviewer #1: Yes

Reviewer #2: Yes

4. Have the authors made all data underlying the findings in their manuscript fully available?

Reviewer #1: Yes

Reviewer #2: Yes

5. Is the manuscript presented in an intelligible fashion and written in standard English?

Reviewer #1: Yes

Reviewer #2: Yes

6. Review Comments to the Author

Reviewer #1: After reviewing the revised manuscript and the authors’ detailed responses, I confirm that all major and minor comments have been thoroughly addressed. The authors substantially improved the discussion of the missing TT genotype, supported by appropriate references and a clearer explanation of selective breeding and allele frequency dynamics. The inclusion of a dedicated Limitations section strengthens the transparency of the study, particularly regarding sample size and the constraints of the TaqMan assay.

The expanded discussion on the functional relevance of the SCD gene and its role in fatty acid metabolism enhances the biological interpretation of the findings. Minor issues related to terminology, reference formatting, figure integration, and abbreviation consistency have been adequately corrected.

Overall, the manuscript has improved significantly in clarity, scientific rigor, and presentation. I recommend acceptance after minor editorial proofreading.

Reviewer #2: Most of the previous comments are correctly addressed in the manuscript, although introductory lines of the abstract still seems lengthy.

RESPONSE: We are grateful to both reviewers for their second round comments. We have made some minor editorial proofreading changes and cut back on the lengthy introductory lines.

---

## [Decision Letter · Decision Letter 2]

9 Dec 2025

Validation of stearoyl-coA desaturase gene TaqMan probe-based SNP for genotyping Tattykeel Australian White MARGRA lamb for health-beneficial omega-3 long-chain fatty acids and intramuscular fat content

PONE-D-25-49554R2

Dear Dr. Malau-Aduli,

We’re pleased to inform you that your manuscript has been judged scientifically suitable for publication and will be formally accepted for publication once it meets all outstanding technical requirements.

Kind regards,

Ayman A Swelum

Academic Editor

PLOS One

Additional Editor Comments (optional):

Reviewers' comments:

Reviewer's Responses to Questions

**Comments to the Author**

Reviewer #2: All comments have been addressed

2. Is the manuscript technically sound, and do the data support the conclusions?

Reviewer #2: Yes

3. Has the statistical analysis been performed appropriately and rigorously?

Reviewer #2: Yes

4. Have the authors made all data underlying the findings in their manuscript fully available?

Reviewer #2: Yes

5. Is the manuscript presented in an intelligible fashion and written in standard English?

Reviewer #2: Yes

Reviewer #2: (No Response)

**Do you want your identity to be public for this peer review?** For information about this choice, including consent withdrawal, please see our Privacy Policy

Reviewer #2: No

---

## [Editor Report · Acceptance letter]

PONE-D-25-49554R2

PLOS One

Dear Dr. Malau-Aduli,

I'm pleased to inform you that your manuscript has been deemed suitable for publication in PLOS One. Congratulations! Your manuscript is now being handed over to our production team.

Kind regards,

on behalf of

Professor Ayman A Swelum

Academic Editor

PLOS One